# Phosphorylation of CaMK and CREB-Mediated Cardiac Aldosterone Synthesis Induced by Arginine Vasopressin in Rats with Myocardial Infarction

**DOI:** 10.3390/ijms232315061

**Published:** 2022-12-01

**Authors:** Yuan-Sheng Zhai, Jie Li, Longyun Peng, Guihua Lu, Xiuren Gao

**Affiliations:** 1Department of Cardiology, The First Affiliated Hospital, Sun Yat-sen University, Guangzhou 510700, China; 2Key Laboratory on Assisted Circulation, Ministry of Health, Guangzhou 510700, China

**Keywords:** aldosterone, arginine vasopressin, myocardial infarction, fibrosis, heart failure, the adrenal cortex

## Abstract

Both aldosterone and arginine vasopressin (AVP) are produced in the heart and may participate in cardiac fibrosis. However, their relationship remains unknown. This study aims to demonstrate the regulation and role of AVP in aldosterone synthesis in the heart. Rats were subjected to a sham operation or myocardial infarction (MI) by ligating the coronary artery. Cardiac function and fibrosis were assessed using echocardiography and immunohistochemical staining, respectively. In addition, the effects of AVP stimulation on cardiac microvascular endothelial cells (CMECs) were studied using ELISA, real-time PCR, and Western blotting. Compared with the rats having undergone a sham operation, the MI rats had an increased LVMI, type I collagen composition, and concentrations of aldosterone and AVP in the heart but decreased cardiac function. As the MI rats aged, the LVMI, type I collagen, aldosterone, and AVP increased, while the LVMI decreased. Furthermore, AVP time-dependently induced aldosterone secretion and CYP11B2 mRNA expression in CMECs. The p-CREB levels were significantly increased by AVP. Nevertheless, these effects were completely blocked by SR49059 or partially inhibited by KN93. This study demonstrated that AVP could induce the secretion of local cardiac aldosterone, which may involve CaMK and CREB phosphorylation and CYP11B2 upregulation through V1 receptor activation.

## 1. Introduction

Acute myocardial infarction (AMI) remains the leading cause of heart failure (HF) worldwide. Recent research reported that about 28% of 86,771 patients with AMI developed HF either during hospitalization for the incident myocardial infarction (MI) or shortly after discharge [1]. Consequently, it is significant to uncover the underlying mechanisms involved in MI-related HF.

Cardiac fibrosis, the accumulation of extracellular matrix (ECM) proteins in the cardiac interstitium, is a reparative or maladaptive process in a variety of myocardial conditions, such as increased hemodynamic load and cardiomyocyte loss [2]. Thus, the progress of cardiac fibrosis may depend on the interaction between profibrotic and antifibrotic pathways [3]. It has been demonstrated that aldosterone can promote hypertension-induced fibrosis through the induction of an inflammatory profibrotic phenotype and the inhibition of antifibrotic factors. In addition, increased aldosterone may be involved in post-MI cardiac fibrosis, which is well-established as the most important pathophysiological basis for MI-related HF [4]. It is widely acknowledged that aldosterone causes cardiac fibrosis not only via the effects of circulating aldosterone produced in the adrenal glands but also through the local synthesis in the heart. As a steroid hormone, aldosterone is primarily synthesized in the zona glomerulosa of the adrenal cortex. An increasing amount of evidence has supported the existence of an “aldosterone tissular system” in the heart [5,6]. Increased levels of aldosterone and the aldosterone synthase gene, CYP11B2, were detected in the non-infarcted left ventricular myocardium of MI rats although the levels of the plasma aldosterone remained unchanged [4]. It is widely acknowledged that aldosterone synthase, the rate-limiting enzyme of aldosterone production, is encoded by CYP11B2 in the zona glomerulosa of the adrenal glands. Mary et al. found that the angiotensin-II-induced expression of human CYP11B2 was partially inhibited by calmodulin kinase (CaMK) inhibitor KN93 or the mutation of cyclic adenosine monophosphate (cAMP)-response element-binding (CREB) protein in adrenal cells, whereby suggesting that CaMK and CREB may participate in the regulation of human CYP11B2 expression [7]. Nevertheless, until recently, little progress has been made in the intracellular mechanisms involved in regulating local cardiac aldosterone synthesis.

As a neurohypophysial peptide, arginine vasopressin (AVP) is predominantly synthesized in the paraventricular and supraoptic nuclei of the hypothalamus and then stored in the posterior pituitary. A growing number of studies have attested that AVP could result in the deterioration of HF through water retention [8]. Since AVP has been reported to be expressed in the endothelial cells, vascular smooth muscle cells (VSMCs), and perivascular tissue in isolated rat hearts, we hypothesized that AVP might be independently synthesized in the heart and exert detrimental effects via the autocrine and paracrine pathways [9].

It is established that most AVP’s actions related to HF are mediated by the V1 receptor (V1R), distributed in cardiac myocytes, VSMCs, endothelial cells, and fibroblasts [10]. Our previous study demonstrated the distribution of AVP in rat cardiac microvascular endothelial cells (CMECs) [11]. AVP stimulation led to the proliferation of cardiac fibroblasts via the V1R, which may be related to cardiac fibrosis in MI rats. Interestingly, AVP and aldosterone were simultaneously increased in the rats with MI. However, the exact relationships between AVP and aldosterone in the heart have yet to be illustrated.

It is well-established that aldosterone causes cardiac fibrosis primarily by activating the mineralocorticoid receptor (MR), a member of the steroid nuclear receptor family [12]. Clinical trials demonstrated that the mineralocorticoid receptor antagonist (MRA) decreased morbidity and mortality in patients with left ventricular systolic dysfunction by inhibiting aldosterone-induced cardiac fibrosis [13,14]. However, recent studies have found that aldosterone also initiates intracellular second messenger cascades and leads to the proliferation of cardiac fibroblasts by activating their transmembrane receptors, which cannot be blocked by MRAs [15]. In addition, hypotension and renal insufficiency limit MRAs’ clinical utility in end-stage HF patients, indicating that suppressing local cardiac aldosterone synthesis may be an alternative to MRAs for HF patients. Since AVP has been found to promote the secretion of aldosterone in the adrenal glands, we sought to determine whether AVP participates in the synthesis of aldosterone in the heart [16].

Against this background, we performed the present study to explore whether AVP and aldosterone are involved in MI-induced cardiac fibrosis and whether AVP participates in regulating the local cardiac aldosterone synthetic system.

## 2. Results

### 2.1. Left Ventricular Mass Index and Cardiac Function in Rats with MI

The left ventricular mass index (LVMI) was greater in the MI rats than in the rats having undergone a sham operation, all of which were of the same age. In addition, the LVMI increased with age among the MI groups. Compared with the sham groups, the MI groups had an increased left ventricular end-systolic diameter (LVESD) and left ventricular end-diastolic diameter (LVEDD) but a decreased left ventricular ejection fraction (LVEF). As the MI rats aged, they had a greater LVESD and LVEDD but a lower LVEF, thus suggesting a deterioration of cardiac function with the increase in age (Table 1 and Appendix A).

### 2.2. Type I Collagen Expression in Rats with MI

As depicted in Figure 1A, a light microscope revealed that myocardial cells were blue and that type I collagen (collagen I) was brown (Figure 1A). Compared with the rats having undergone a sham operation, a marked deposition of collagen I was detected in the left ventricular non-infarcted interstitial tissues of the MI rats of the same age. Moreover, with the increase in age, the protein expressions of collagen I were significantly elevated among the rats with MI, which peaked at the age of 16 weeks post-infarction. (Figure 1B).

### 2.3. Aldosterone and AVP Concentrations in Rats with MI

As shown in Figure 2A,B, the concentrations of aldosterone and AVP increased in the MI rats in comparison with the rats having undergone the sham operation. As the MI rats aged, both aldosterone and AVP significantly increased, reaching their highest level at the age of 16 weeks post-infarction (Figure 2A,B). However, we observed comparable levels of aldosterone and AVP among the rats having undergone the sham operation. Correlation analyses showed that there was a positive correlation between cardiac AVP and aldosterone (Figure 2C). Moreover, either AVP or aldosterone was positively associated with collagen I expression but inversely with the LVEF (Appendix A).

### 2.4. Effects of AVP on Aldosterone Secretion in CMECs

First, we treated the CMECs at various concentrations of AVP. As displayed in Figure 3A, 10^−6^ mol/L AVP contributed to more aldosterone secretion than 10^−7^ mol/L AVP in CMECs. By contrast, 10^−5^ mol/L or 10^−8^ mol/L AVP did not affect CMECs. Then, we found that the aldosterone secretion was time-independently induced after CMECs were stimulated with 10^−7^ mol/L AVP for 12, 24, and 48 h, respectively (Figure 3B). After that, CMECs were treated with AVP (10^−7^ mol/L) alone or in combination with AVP V1R antagonist (SR49059, Sigma, St Louis, MO, USA) or CaMK antagonist (KN93, Sigma, St Louis, MO, USA) for 24 h. In comparison with the vehicle, AVP significantly increased aldosterone production in CMECs. These effects were completely blocked by SR49059 or partially inhibited by KN93 (Figure 3C).

### 2.5. Effects of AVP on CYP11B2 mRNA Expression in CMECs

Compared with the vehicle, 10^−6^ mol/L or 10^−7^ mol/L AVP resulted in significantly higher CYP11B2 mRNA expression, whereas 10^−5^ mol/L or 10^−8^ mol/L AVP demonstrated comparable results. Similar to the aldosterone secretion induced by AVP, 10^−6^ mol/L AVP had the highest CYP11B2 mRNA expression in CMECs (Figure 4A). Afterward, CYP11B2 mRNA expression was time-independently upregulated when 10^−7^ mol/L AVP was administered to CMECs for 12, 24, and 48 h, respectively (Figure 4B). Furthermore, when CMECs were stimulated with AVP (10^−7^ mol/L) alone or in combination with SR49059 (10^−6^ mol/L) or KN93 (10^−6^ mol/L) for 24 h, AVP-upregulated CYP11B2 mRNA expression was completely suppressed by SR49059 or partially by KN93 (Figure 4C).

### 2.6. Effects of AVP on Protein Expression of Phosphorylated CREB in CMECs

As depicted in Figure 5, AVP (10^−7^ mol/L) significantly upregulated the protein expression of phosphorylated CREB (p-CREB) in comparison with the vehicle. However, these effects were completely abolished when SR49059 (10^−6^ mol/L) was added to the cell culture medium before AVP. Furthermore, we observed that the AVP-induced upregulation of p-CREB was partially inhibited by KN93 (10^−6^ mol/L) (Figure 5).

## 3. Discussion

In this study, three key findings were revealed: (1) both AVP and aldosterone were involved in cardiac fibrosis and heart failure in post-infarction rats; (2) AVP was able to induce aldosterone secretion in CMECs by enhancing the expression of CYP11B2; and (3) AVP upregulated CYP11B2 expression through promoting the phosphorylation of CaMK and CREB after binding to the V1R. These intracellular regulatory mechanisms of AVP in aldosterone synthesis in the local heart may have an important implication in clinical medicine.

Cardiac interstitial fibrosis, characterized by the production and deposition of ECM proteins, is a compensatory response secondary to AMI [2]. Moderate cardiac fibrosis is beneficial to repair a post-infarction heart, but excessive fibrosis may have harmful effects on cardiac function and eventually lead to HF. It has been demonstrated that collagen I accounts for 80% of the total collagen newly synthesized from fibroblasts in the heart [17]. Thus, an increase in collagen I represents the deposition of the cardiac interstitium. Our study showed that collagen I expression increased with age in post-infarction rats and caused ventricular remodeling, supported by an increase in LVMI. Conversely, the LVEF, an index of cardiac systolic function, was age-dependently reduced. There was a negative correlation between collagen I expression and LVEF. These results implied that an MI-induced increase in collagen I expression might contribute to the deterioration of cardiac function.

Previous studies have demonstrated that aldosterone is synthesized not only in the zona glomerulosa of the adrenal cortex but also in the heart, as evidenced by the existence of aldosterone synthase in cardiomyocytes, VSMCs, and endothelial cells [18,19]. Although the heart produces less aldosterone than the adrenal glands, the aldosterone concentration in the heart is 17 times higher than the plasma in non-adrenalectomized animals, indicating that cardiogenic aldosterone may specifically act on the heart [6]. Consistent with prior studies, the present study revealed that aldosterone concentrations were age-dependently increased in the left ventricular non-infarcted region of post-infarction rats and positively associated with collagen I, which suggests that aldosterone may lead to cardiac fibrosis. These aldosterone-induced effects might relate to the activation of MRs, which prevail in cardiomyocytes, cardiac fibroblasts, endothelial cells, and VSMCs [20]. There are three possible mechanisms of aldosterone-induced cardiac fibrosis. First, aldosterone could promote fibroblast differentiation to myofibroblasts, marked by the increased expression of α-smooth muscle actin (αSMA), and contributed to the elevated production of collagens [21]. In addition, aldosterone may enhance collagen I synthesis after activating the MR-dependent pathway in VSMCs, which was able to be blocked by spironolactone [22]. Furthermore, it was suggested that cardiac fibrosis might be associated with angiogenesis [23]. Eplerenone, an MR antagonist, was able to attenuate cardiac fibrosis by inhibiting angiogenesis, indicating that aldosterone may be involved in angiogenesis. However, another study showed that aldosterone significantly reduced angiogenesis, which was prevented by MR deletion [24]. Consequently, the relationship between aldosterone, angiogenesis, and cardiac fibrosis should be further investigated.

Angiotensin II is the major stimulator of aldosterone in the adrenal glands. However, with the discovery of “aldosterone escape”, it is suggested that other hormones, such as AVP, adrenocorticotropic hormone, and catecholamine, can promote aldosterone secretion in the adrenal glands as well [25,26]. Similarly, the production of local cardiac aldosterone has been revealed to be modulated primarily through angiotensin II and could be blunted by the AT1 receptor antagonists. Nevertheless, whether cardiac aldosterone synthesis is influenced by other hormones, including AVP, remains unclear. Our study showed that the concentrations of AVP were increased and positively associated with collagen I expression. Interestingly, there was also a positive association between aldosterone and AVP, implying that both may be involved in cardiac fibrosis. However, it is challenging for in vivo studies to determine how AVP and aldosterone interact with each other.

After that, we used CMECs to determine the direct effects of AVP on aldosterone biosynthesis. The results showed that AVP time-independently promoted CYP11B2 mRNA expression and aldosterone secretion. Surprisingly, we did not observe the upregulation of CYP11B2 mRNA expression to be proportional to the concentration of AVP. The exact mechanism remains unknown; however, it may be related to CMEC apoptosis or the death induced by an overdose of AVP. Aldosterone synthase, a cytochrome P-450 enzyme encoded by the gene CYP11B2, generated aldosterone from 11-deoxycorticosterone, the final step of synthetic pathways [27]. Angiotensin II and K^+^ were the primary regulators of aldosterone synthase and CYP11B2 in the adrenal glands [28]. Two cis-elements were identified to be necessary for the CYP11B2 transcription induced by angiotensin II and K^+^: one was the cAMP-response element, and the other was an element binding an orphan nuclear receptor, i.e., steroidogenic factor-1 (SF-1), or a second orphan nuclear receptor, i.e., chicken ovalbumin upstream promoter transcription factor (COUP-TF) [29]. Deleting these elements from the human CYP11B2 5′-flanking region led to reduced basal levels of transcription and maximal induction by cAMP. Furthermore, the cAMP-response element enhanced the transcription of CYP11B2 induced by angiotensin II through a calcium-signaling pathway, supported by the fact that the phosphorylation of CREB by CaMK I and CaMK IV could increase CYP11B2 gene transcription [7,30,31]. By contrast, CaMK I-induced CYP11B2 reporter activity was inhibited when the cAMP-response element suffered from mutation. However, there are few data on regulating the CYP11B2 gene transcription in the heart. Our study revealed that AVP significantly upregulated the expression of p-CREB in CMECs, which could be partially blocked by KN93, an antagonist of CaMK. These results indicated that the regulatory mechanism of CYP11B2 gene transcription in the heart might be analogous to the adrenal glands, and AVP enhanced CYP11B2 gene transcription through the phosphorylation of CaMK and CREB. As CYP11B2 gene transcription could only be partly repressed by KN93, we believe some other intracellular pathways might be participating in the synthesis of the aldosterone induced by AVP, such as the inducible nitric oxide synthase (iNOS)–nitric oxide (NO) systems. As AVP-induced NO was able to offset the effects of AVP on cardiac fibroblasts, it is interesting to explore whether iNOS–NO systems participate in AVP-induced aldosterone synthesis in CMECs, which could also produce NO [32].

Three receptors mediate AVP action in fluid and cardiovascular homeostasis, including the V1R, the V2 receptor, and the V3 receptor [10]. The V1R, primarily located in blood vessels, has also been found in the heart. Activating the V1R could immediately elicit arteriolar and coronary vasoconstriction and eventually lead to the synthesis of the proteins involved in cellular hypertrophy with sustained AVP stimulation [9,33]. It was recently suggested that the V1R might also facilitate the action of AVP in cardiac fibroblasts proliferation [34]. In agreement with previous studies, the V1R could mediate the upregulation of the CYP11B2 expression induced by AVP in the present study. This was evidenced by the fact that SR49059 could completely block the aforementioned effects. In addition, we found that SR49059 could reduce the expression of p-CREB. As a member of G-protein-coupled receptors, the V1R increased the intracellular calcium concentration through the influx of extracellular Ca^2+^ and the mobilization of intracellular Ca^2+^. The increase in intracellular calcium may contribute to the parallel activation of several pathways, such as calcium/CaMK, protein kinase C (PKC), and p42/p44 mitogen-activated protein kinase (MAPK) [35]. Based on the results of our study, it is plausible to speculate that the activation of the V1R enhanced CYP11B2 gene transcription via the activation of calcium/CaMK and the phosphorylation of CREB in our study.

In conclusion, this study confirmed the existence of the aldosterone synthesis system in the heart. AVP could independently promote the secretion of aldosterone, which may contribute to MI-induced cardiac fibrosis. The cellular mechanism of the effect of AVP on aldosterone production may involve CaMK and CREB phosphorylation through V1R activation. Clinical studies should be performed to evaluate these beneficial effects of V1R in MI patients.

## 4. Materials and Methods

### 4.1. Animals

Male Sprague–Dawley rats weighing 250–350 g were purchased from the Medical Experimental Animal Center of Guangdong Province (Guangzhou, China). Standard chow and water were available ad libitum, and the animals were kept in a constant-temperature environment with 12 h light-dark cycles.

### 4.2. Experimental Protocol

As previously described, a rat model of MI was generated by ligating the left anterior descending coronary artery. After coronary artery ligation, 32 surviving rats were randomly assigned to three groups: (1) the 6-week post-infarction group, which was bred for 6 weeks after surgery (6w MI, *n* = 10); (2) the 11-weeks post-infarction group (11w MI, *n* = 11); and (3) the 16-week post-infarction group (16w MI, *n* = 11). Thirty rats were subjected to a sham operation and randomly divided into three groups: (1) the 6-week post-sham-operation group, which was fed for 6 weeks after surgery (6w sham, *n* = 10); (2) the 11-week post-sham-operation group (11w sham, *n* = 10); and (3) the 16-week post-sham-operation group (16w sham, *n* = 10).

At the end of the 16-week study, there were 10, 10, 10, 9, 9, and 8 surviving rats in the groups of 6w sham, 11w sham, 16w sham, 6w MI, 11w MI, and 16w MI groups, respectively. The data from the dead rats were not included in the analyses.

Before the experimental rats were sacrificed, echocardiography was used to determine their cardiac function. After being anesthetized via an intraperitoneal injection of 10% chloralhydrate (3 mL/kg), their hearts were excised and immediately placed in an ice-cold NaCl 0.9% buffer solution. Afterward, the right ventricular free wall and atria were separated from the left ventricles and the septum, which were then rinsed with isotonic saline. After that, the dried left ventricles and the septum were weighed and dissected into the apex, middle ring, and base. The transverse sections of the heart were either placed in 10% neutral formalin for immunohistochemical detection or frozen in liquid nitrogen for further analysis.

LVMI, the ratio of left ventricular mass to body weight, was calculated and used as an index of ventricular remodeling.

The study was conducted strictly following the guidelines of the Sun Yat-Sen University Institutional Animal Care and Use Committee, and the various procedures were approved by the Experimental Animal Ethics Committee of the Sun Yat-Sen University (Guangzhou, China, SYSU-IACUC-2018-000281).

### 4.3. Cardiac Function Assessment via Echocardiography

To assess the cardiac function of the rats, transthoracic echocardiography was performed by an experienced operator blinded to the group allocation. After the rats were anesthetized by inhaling 4.0% isoflurane, echocardiography was performed using a Vevo2100 ultrasound machine equipped with a 30 MHz transducer (VisualSonics, Inc., Toronto, Ontario, Canada). The M-mode tracings of the long-axis view of the left ventricle were obtained, and the following parameters were measured: LVESD and LVEDD. LVEF, the index of cardiac systolic function, was calculated. All the parameters were measured from three consecutive cardiac cycles, and the average was used for data analyses.

### 4.4. Immunohistochemical Analysis of Type I Collagen

Since cardiac fibrosis contributes to the development of MI-induced HF, we examined whether interstitial collagen was deposited in the non-infarcted region of the left ventricle. Immunohistochemical staining was used to detect collagen I expression in the left ventricular non-infarcted region (as previously described) [36]. Leaving out the primary antibody served as a negative control to ensure the specificity of the antibody. Sections were observed with 200× magnification, and the integrated optical density of the collagen was measured using the Image J Analytical System (National Institutes of Health, Bethesda, MD, USA). Five random fields were analyzed for each section to yield a single number. The results were presented as average optical density (AOD), which was calculated by the integrated optical density of collagen I positive staining divided by the positive area.

### 4.5. Cardiac Microvascular Endothelial Cell Culture In Vitro

Primary CMECs were obtained from iCell Bioscience, Inc. (Shanghai, China). CMECs were cultured in Dulbecco’s modified Eagle’s medium/F12 (DMEM/F12, Gibco, Carlsbad, CA, USA) containing 1% penicillin/streptomycin and 10% fetal bovine serum (FBS, Gibco, Carlsbad, CA, USA) and incubated in a humidified atmosphere of 5% CO_2_ at 37 °C. The CMECs in the third and fourth passages were used for the study. After starvation for 24 h, the CMECs were treated with AVP, SR49059, KN93, or a vehicle for at least 12 h.

### 4.6. Enzyme-Linked Immunosorbent Assay

To elucidate the relationship between aldosterone and AVP, we tested their concentrations in the myocardium tissues of the left ventricular non-infarcted region or in the cell culture medium of CMECs using an enzyme-linked immunosorbent assay (ELISA). The myocardial tissues from the non-infarcted region of the left ventricle were grounded with a glass homogenizer in a phosphate-buffered saline solution (0.01 M, pH 7.4) and centrifuged at 3000 r/min for 30 min. The supernatant of homogenized myocardial tissues was collected to detect AVP and aldosterone, while the cell culture medium of CMECs was harvested to measure aldosterone levels. Subsequently, the protein concentrations of AVP and aldosterone were examined using commercially obtained ELISA kits (BlueGene Biotech Co., Shanghai, China) following the manufacturer’s instructions.

### 4.7. Real-Time Polymerase Chain Reaction

To determine the mechanism through which AVP regulated the synthesis of aldosterone in CMECs, we detected the mRNA expression of CYP11B2 using real-time polymerase chain reaction (RT-PCR) as previously described [37]. The total RNA was extracted from the cell culture medium of CMECs with a TRIzol Reagent (Sigma, St Louis, MO, USA) and subjected to RT-PCR analysis via an RT kit (Toyobo Co., Osaka, Japan), following the manufacturer’s instructions. The amplified PCR products were detected using fluorescence staining with SYBR Green (Vazyme Biotech Co., Nanjing, China). The forward and reverse primers for rat CYP11B2 and actin are presented in Table 2. Actin, an endogenous control gene, was used to normalize the relative mRNA expression of CYP11B2.

### 4.8. Western Blotting

To demonstrate the intracellular regulation of AVP-induced CYP11B2 expression, Western blotting was performed to determine the protein levels of p-CREB in CMECs (as described previously) [36]. The primary antibodies were rabbit monoclonal antibodies from Cell Signaling Technology (Danvers, MA, USA) at a 1:500 dilution and rabbit polyclonal anti-GAPDH antibodies from Servicebio Technology Co. (Wuhan, China) at a 1:10,000 dilution. The secondary antibody was the goat anti-rabbit horseradish peroxidase-conjugated secondary antibody from Proteintech at a 1:10,000 dilution. The bands were visualized using an enhanced chemiluminescence kit (Cell Signaling Technology, Danvers, MA, USA) and analyzed using the Image J Analytical System. GAPDH was used to normalize the data of p-CREB, which were expressed by the optical density ratio of the treatment groups to the vehicle group.

### 4.9. Statistical Analysis

Statistical analyses were performed using the SPSS software (version 23.0). The continuous variables are presented as the mean ± SD when they were normally distributed or as median and interquartile range when they did not meet a normal distribution. The categorical variables are expressed as proportions. The normality and variance homogeneity of the data were checked by using Shapiro–Wilk and Levene tests, respectively. A comparison of the continuous variables between the two groups was performed using a two-tailed Student’s *t*-test or a non-parametric test, depending on the normality of the distribution. The categorical variables were analyzed with chi-square tests. Multiple-group differences were analyzed using the one-way analysis of variance (ANOVA) followed by a post hoc test or the Kruskal–Wallis test followed by Bonferroni analysis. The correlations between the two groups were assessed using Pearson’s chi-square test. *p* < 0.05 was considered to be statistically significant.

## Figures and Tables

**Figure 1 ijms-23-15061-f001:**
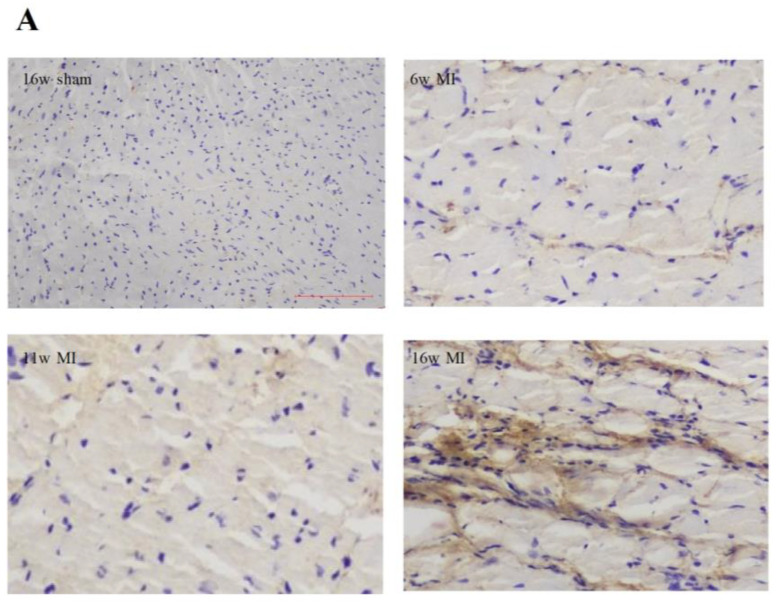
Type I collagen expression in left ventricular non-infarcted region of rats. MI, myocardial infarction; sham, rats having undergone a sham operation; 16w sham, 16-week post-sham-operation group; 6w MI, 6-week post-infarction group; 11w MI, 11-week post-infarction group; 16w-MI, 16-week post-infarction group. (**A**) Representative images of immunohistochemical staining for type I collagen (collagen I) (magnification: ×200). Blue indicates nucleus, and brown indicates collagen I. (**B**) Graphical representation of collagens I protein expression in the sections. AOD, average optical density; the results are presented as mean ± SD. ^#^
*p* < 0.05 vs. sham-operated rats of the same age; * *p* < 0.05 vs. 6w MI; ^+^
*p* < 0.05 vs. 11w MI.

**Figure 2 ijms-23-15061-f002:**
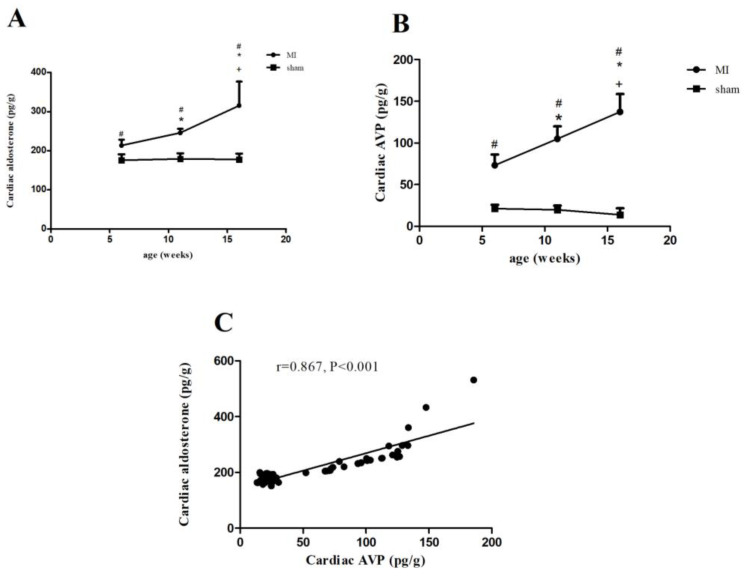
Concentrations of aldosterone and AVP in left ventricular non-infarcted region of rats. (**A**) Representative graph of aldosterone concentration. (**B**) Representative graph of AVP concentration. AVP, arginine vasopressin. (**C**) Correlation analysis between aldosterone and AVP. ^#^
*p* < 0.05 vs. sham-operated rats of the same age; * *p* < 0.05 vs. 6w MI; ^+^
*p* < 0.05 vs. 11w MI.

**Figure 3 ijms-23-15061-f003:**
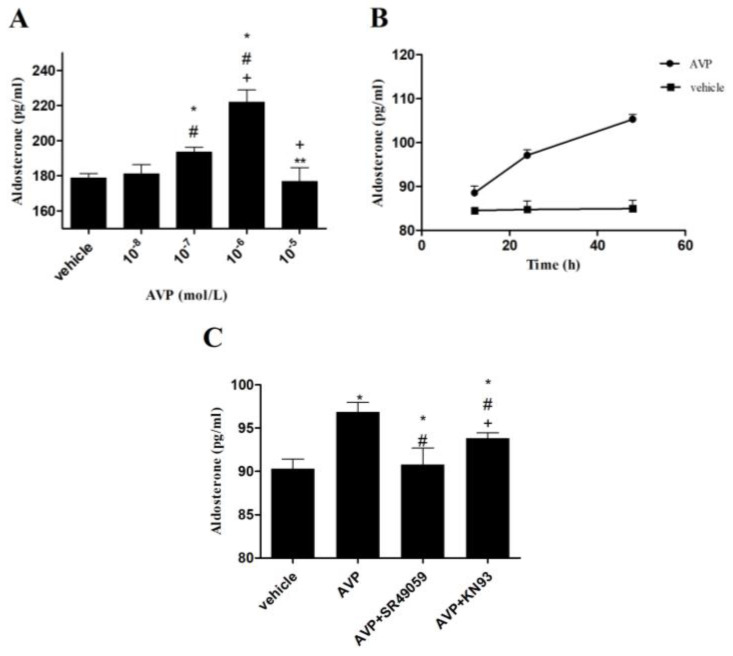
Effects of AVP on aldosterone secretion in CMECs. CMECs, cardiac microvascular endothelial cells; SR49059, AVP V1R antagonist; KN93, CaMK antagonist. (**A**) Effects of different concentrations of AVP on aldosterone secretion. * *p* < 0.05 vs. vehicle; ^#^
*p* < 0.05 vs. 10^−8^ mol/L AVP; ^+^
*p* < 0.05 vs. 10^−7^ mol/L AVP; ** *p* < 0.05 vs. 10^−6^ mol/L AVP. (**B**) AVP (10^−7^ mol/L) time-independently induced secretion of aldosterone. (**C**) Effects of different antagonists on AVP-induced secretion of aldosterone. AVP, 10^−7^ mol/L; SR49059, 10^−6^ mol/L; KN93, 10^−7^ mol/L. * *p* < 0.05 vs. vehicle; ^#^
*p* < 0.05 vs. AVP; ^+^
*p* < 0.05 vs. (AVP + SR49059).

**Figure 4 ijms-23-15061-f004:**
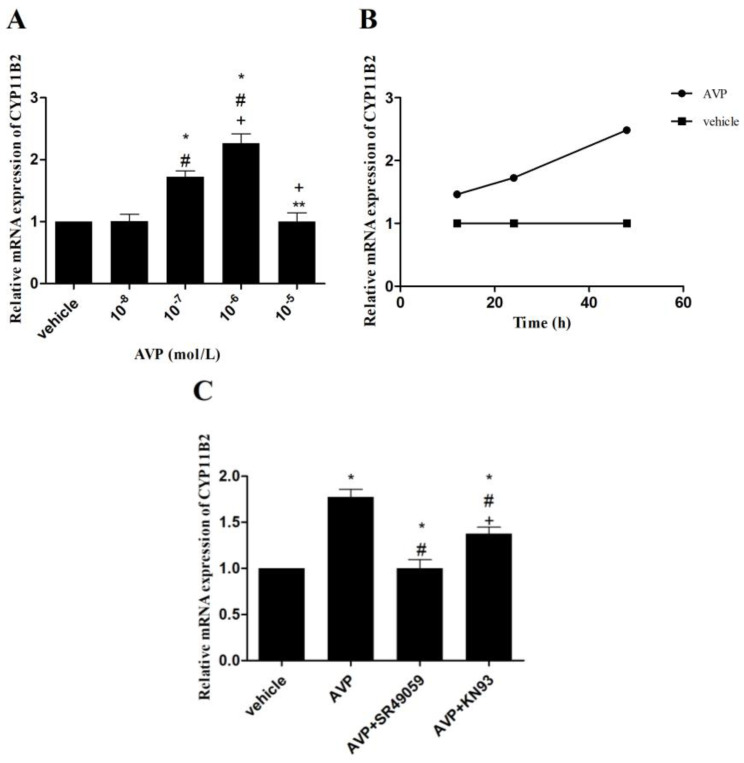
Effects of AVP on CYP11B2 mRNA expression in CMECs. (**A**) Effects of different concentrations of AVP on CYP11B2 mRNA expression. * *p* < 0.05 vs. vehicle; ^#^
*p* < 0.05 vs. 10^−8^ mol/L AVP; ^+^
*p* < 0.05 vs. 10^−7^ mol/L AVP; ** *p* < 0.05 vs. 10^−6^ mol/L AVP. (**B**) AVP (10^−7^ mol/L) time-independently upregulated CYP11B2 mRNA expression. (**C**) Effects of different antagonists on AVP-upregulated CYP11B2 mRNA expression. AVP, 10^−7^ mol/L; SR49059, 10^−6^ mol/L; KN93, 10^−6^ mol/L. * *p* < 0.05 vs. vehicle; ^#^
*p* < 0.05 vs. AVP; ^+^
*p* < 0.05 vs. (AVP + SR49059).

**Figure 5 ijms-23-15061-f005:**
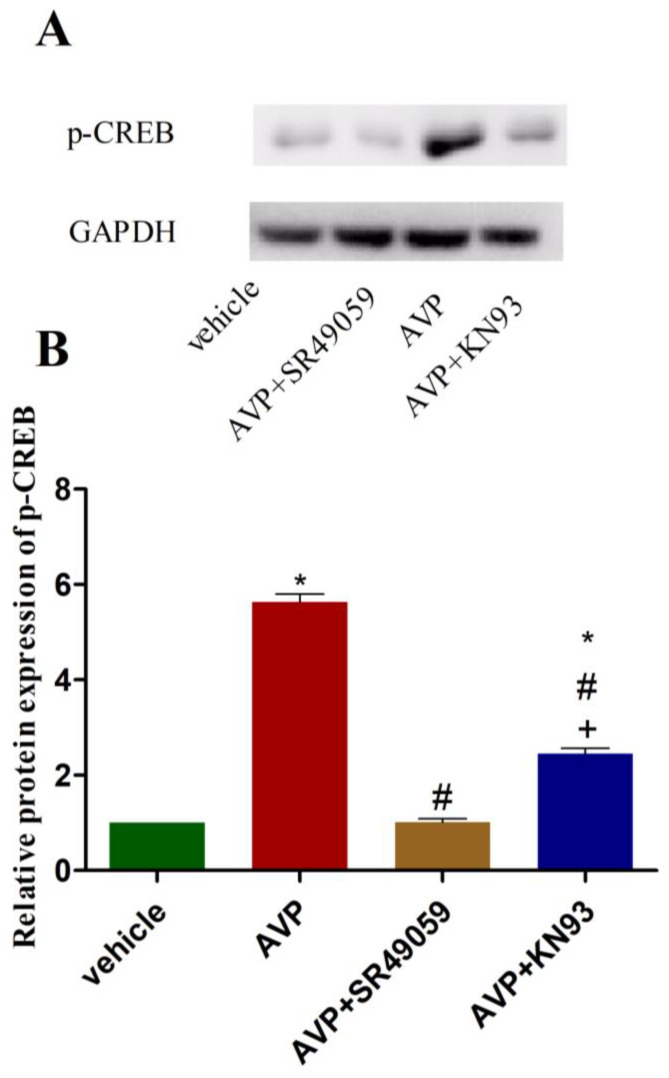
Effects of AVP on p-CREB expression in CMECs. p-CREB, phosphorylated CREB; AVP, 10^−7^ mol/L; SR49059, 10^−6^ mol/L; KN93, 10^−6^ mol/L. (**A**) Representative images of Western blots. (**B**) Graphical representation of protein expression of p-CREB. * *p* < 0.05 vs. vehicle; ^#^
*p* < 0.05 vs. AVP; ^+^
*p* < 0.05 vs. (AVP + SR49059).

**Table 1 ijms-23-15061-t001:** Left ventricular mass index and cardiac function.

Parameters	6w Sham	11w Sham	16w Sham	6w MI	11w MI	16w MI
	*n* = 10	*n* = 10	*n* = 10	*n* = 9	*n* = 9	*n* = 8
BW (g)	414 ± 17	504 ± 28	514 ± 24	406 ± 16	497 ± 43 ^b^	480 ± 16 ^ab^
LVM (mg)	773 ± 45	950 ± 86	968 ± 82	833 ± 67 ^a^	1101 ± 119 ^ab^	1212 ± 74 ^abc^
LVMI (mg/g)	1.87 ± 0.1	1.88 ± 0.1	1.9 ± 0.1	2.1 ± 0.1 ^a^	2.2 ± 0.1 ^ab^	2.5 ± 0.1 ^abc^
LVESD (mm)	5.42 ± 1.33	5.54 ± 0.45	5.15 ± 1.08	6.97 ± 0.88 ^a^	7.92 ± 0.67 ^ab^	10.05 ± 1.08 ^abc^
LVEDD (mm)	7.72 ± 1.17	7.76 ± 0.54	7.45 ± 1.06	8.73 ± 0.92 ^a^	9.34 ± 75.97 ^a^	11.18 ± 1.19 ^abc^
LVEF (%)	55.6 ± 12.0	53.2 ± 5.63	57.6 ± 8.3	39.7 ± 5.2 ^a^	30.7 ± 4.0 ^ab^	20.8 ± 2.5 ^abc^

BW, body weight; LVM, left ventricular mass; LVMI, left ventricular mass index; LVESD, left ventricular end-systolic diameter; LVEDD, left ventricular end-diastolic diameter; LVEF, left ventricular ejection fraction; 6w sham, 6-week post-sham-operation group; 11w sham, 11-week post-sham-operation group; 16w sham, 16-week post-sham-operation group; 6w MI, 6-week post-infarction group; 11w MI, 11-week post-infarction group; 16w MI, 16-week post-infarction group. Data are presented as mean ± SD. ^a^
*p* < 0.05 vs. sham-operated rats of the same age; ^b^
*p* < 0.05 vs. 6w MI; ^c^
*p* < 0.05 vs. 11w MI.

**Table 2 ijms-23-15061-t002:** Primers used for the PCR analysis.

Genes	Forward Sequence	Reverse Sequence
Actin	5′-CGTTGACATCCGTAAAGACCTC-3′	5′-TAGGAGCCAGGGCAGTAATCT-3′
CYP11B2	5′-ACCATGGATGTCCAGCAA-3’	5′-GAGAGCTGCCGAGTCTGA-3′

## Data Availability

Not applicable.

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
