# Peer review of "Phosphorylation of CaMK and CREB-Mediated Cardiac Aldosterone Synthesis Induced by Arginine Vasopressin in Rats with Myocardial Infarction"

_ijms, 2022, doi:10.3390/ijms232315061_

Round 1

Reviewer 1 Report

In the manuscript titled “Effects of cardiac arginine vasopressin on aldosterone synthesis in rats with myocardial infarction”, the authors investigated the role of arginine vasopressin in cardiac aldosterone synthesis. The authors were able to demonstrate a possible intracellular cascade regarding aldosterone secretion induced by AVP. This is a novel, and important finding from a translational research and clinical perspective, and provides insight to an intracellular pathway. However, there are some concerns that should kindly be addressed.

Introduction/ Background

Significant language editing is required throughout the manuscript.

Page 1, line34: The authors state that aldosterone is linked to post-MI cardiac fibrosis, yet it is important to note that increased aldosterone is linked to normal, compensatory structural tissue changes following damage/ increased pressure (as the authors correctly refer to in their Discussion (page 8, line 203). However, the development of cardiac fibrosis results from an imbalance between profibrotic and antifibrotic pathways, and aldosterone in itself may worsen fibrosis through possible activation of inflammation/galectin 3-induced fibrosis and inhibition of antifibrotic factors (like BNP and bone morphogenetic protein 4) (Azibani et al, 2012). 

Page 2, line 51: Indeed AVP is a neurohypophysial peptide, and is expressed by endothelial cells and VSMC. Thus the authors might consider the role of the iNOS-NO systems, specifically when referring to cardiac fibroblasts – as this is an endothelial-dependent mechanism. How would aldosterone relate to the iNOS-NO system? As, NO production induced by AVP may counteract the profibrotic effects of AVP, thus the development of myocardial fibrosis also depends on the balance between profibrotic AVP and antifibrotic NO effects on MF (Fan et al 2007). Something to consider when interpreting the background and results.

Results

Page 3, line 106: Did the authors also assess the expression of alpha-smooth-muscle actin (as this is involved in CF myofibroblast transformation? As well as the possible effects of AVP-induced transforming growth factor beta-1 (TGF-beta 1).

Of minor note, have the authors considered the effect of such induced expression, in vivo, as the central AVP may play a role in the degree to which AVP expression in the myocardium is affected?

Discussion

Indeed, the authors found that collagen I expression markedly increased with age, post-infarct, however, have the authors considered the expression of alpha-smooth-muscle actin. as this is involved in CF, to take the additionally functional effects on the VSMC into account?

How would the severity of the infarcted area, affect not only the AVP expression, but the rate of angiotensin synthesis driven by AVP (meant as a theoretical speculative comment to the authors).

Page 9, line 226-227: Indeed, angiotensin may increase the production of collagens, but have the authors considered the affect of angiotensin on neovascularization, (as aldosterone has been shown to enhance ischemia-induced neovascularization).

Page 10, line 268: Although activating the V1R would induce vasoconstriction, the immediate effect will not be LVH – with sustained vasoconstriction/ arterial narrowing etc, structural changes will occur, among other LVH, and thus eventual decreased contractility.

Additional (minor) comments:

Based on the significant findings, the authors might consider rephrasing the manuscript title, to better highlight these key findings. Specifically, that the cellular mechanism of AVP on aldosterone production may involve CaMK and CREB phosphorylation through V1R activation.

Minor esthetical aspect, some of the figure’s axes are labelled in differ fonts - please revise.

What was the reproducibility of the ELISA used and were the echocardiograms done by the same investigator, or various? If so, what was the inter and intra-person variability?

Reviewer 2 Report

Zhai et alt present an intriguing paper titled “Effects of cardiac arginine vasopressin on aldosterone synthesis in rats with myocardial infarction” which demonstrates firstly the relationship between vasopressin and local synthesis of aldosterone in the myocardium, and its relationship with myocardial hypertrophy and potentially with heart failure.

The paper had a rigorous design and the results are strong; however, I’ve got some issues.

  1. Firstly, I found inappropriately set the Methods section after the results and the discussion; it’s against the usual structure of a scientific article. This is not only a formal issue but substantial: the article is hard to read. Moreover, this structure led to multiple repetitions and redundancy between methods and results. The following periods should be rewritten in the correct section:

  • 2.1 “at the end… fraction (LVEF)”, it is substantially absent in methods

  • 2.2 “since… rats with MI”

  • 2.3 “to elucidate…operation”

  • 2.4 “The enzyme-linked… CMECs”

  • 2.5 “to find… results”

  • 2.6 “to demonstrate..blotting”

  1. How did you estimate the number of MI rats and “controls”? It’s important to determine the strength of the results and their reproducibility.

  2. Did you follow some standard guidelines to write the paper? 

  3. The statistical analysis section must be improved. How did you describe categorical data? Did you test normality? You used parametric statistics for two relatively small samples, did you consider non-parametric tests?

  4. Review all the abbreviations: some aren’t explicit, even the universally used ones.

Round 2

Reviewer 2 Report

I found the second version of this article far more readable and less redundant and I appreciate the effort made by the Authors in improving the manuscript in this unusual form. Statistical analisys section is now ok.

If the journal required this organitation of the paper, with results before methods, it's ok for publication.